# RETHINKING MULTIPLE-QUERY OPTIMIZATION FOR APPROXIMATE NEAREST NEIGHBOR SEARCH

## ABSTRACT

Approximate nearest neighbor search (ANNS) over vector databases is a fundamental operation for many modern applications, where rapid processing of queries is of critical importance. In traditional database systems, which face the same requirement, multiple-query optimization (MQO) has been extensively studied to address this challenge. Although MQO is a general technique that exploits shared computation to process a set of queries more efficiently than evaluating each query in isolation, no analogous algorithmic strategy has yet been proposed for ANNS. To this end, we present a novel algorithmic MQO framework tailored to ANNS. The framework is universally applicable to graph-based ANNS methods, delivering significant speedups while keeping both the underlying index and the search algorithm intact. Specifically, we construct a minimum spanning tree over the query vectors and initialize each query's search entry using the nearest neighbor returned by its parent in the tree, thereby revealing and exploiting opportunities for computation reuse. We empirically validate our framework across multiple ANNS methods and datasets, demonstrating its feasibility and effectiveness.

## 1 INTRODUCTION

Given a database $\mathcal{D}$ of vectors, the end goal of approximate nearest neighbor search (ANNS) is to retrieve, for each query vector, its closest neighbor in $\mathcal{D}$ in as little time as possible. This problem serves as a fundamental building block for many modern ML applications, from retrieval-augmented generation (Lewis et al., 2020; Borgeaud et al., 2021) and recommendation systems (Bachrach et al., 2014; Feng et al., 2022) to computer vision tasks (Aiger et al., 2023; Jang & Cho, 2020) and content generation and summarization (Baud & Aussem, 2023). To improve performance, much prior work has focused on devising advanced index structures to accelerate individual query processing (Bentley, 1975; Gionis et al., 1999; Wang et al., 2021; Weber et al., 1998; Malkov & Yashunin, 2018).

In traditional relational databases, which share a similar need for efficient query processing, multiple-query optimization (MQO) introduces a complementary dimension of optimization, offering a different perspective on the same challenge. MQO (Dokeroglu et al., 2014; Bayir et al., 2006; Marcus et al., 2019) identifies and reuses overlapping computations among multiple queries so that collective execution outperforms isolated processing. For relational databases, queries can be decomposed via relational algebra, enabling MQO algorithms to capture shared tasks and avoid redundant work.

Since MQO is a powerful algorithmic concept that can be applied generally, it is natural to ask whether its principles could accelerate query processing in ANNS. Indeed, a batched setting—where multiple queries are submitted to the index simultaneously—commonly arises in ANNS and is supported by many prior works (Jayaram Subramanya et al., 2019; Azizi et al., 2023; Liu et al., 2023; Zhao et al., 2020; Groh et al., 2023; Yu et al., 2022; Ootomo et al., 2024), yet they do not explicitly explore inter-query relationships to uncover reuse opportunities and exploit overlapping computations. To the best of our knowledge, an MQO framework tailored to ANNS has not yet been proposed. This difficulty stems from the fact that ANNS queries are vectors of real numbers, lacking the algebraic structure that makes shared computation explicit in relational settings.

Fortunately, MQO becomes feasible for ANNS once one exploits the inherent structure and details of modern ANNS algorithms. Modern ANNS solutions construct a proximity graph by connecting nearby data points and then perform a greedy search. Starting from an entry point selected according to a predefined policy, the search hops to the neighbor closest to the query until convergence. Because

choosing an entry point near the query reduces the number of hops and hence the search time, initializing the search closer to the query accelerates retrieval. When multiple queries are processed as a batch, we therefore propose using, for each query, the nearest neighbor result of any previously processed query that lies close in the embedding space as its entry point. Through this strategy, we enable indirect reuse of computation by supplying high-quality entry points.

The remaining challenge is to determine a processing strategy that maximizes proximity between each entry point and its corresponding query. We address this by organizing the query batch $Q$ into a minimum spanning tree (MST) based on pairwise distances, which yields a connected structure spanning all queries with minimum total edge length. Each query's search is then initialized at the nearest neighbor result of its parent in the MST, accelerating the overall batch search. We also present a formal theoretical analysis, deriving explicit bounds on the expected search latency to establish provable performance guarantees that underpin our MQO framework.

Furthermore, we introduce a rapid query preprocessing method for scenarios where the full batch is only revealed at runtime and end-to-end latency must be minimized. Constructing the exact MST in such cases is prohibitive, as it requires $\mathcal{O}(|Q|^2)$ pairwise-distance computations. Instead, we partition $Q$ into multiple groups and employ an auxiliary ANNS index to build an approximate MST within each group, reducing computational burden while preserving high-quality processing strategy.

Experimental results show that our framework achieves up to a $1.84\times$ speedup in the search process and a fully end-to-end geometric mean speedup of $1.22\times$. Importantly, these gains are obtained without altering any underlying index structure or search algorithm, and the framework applies effectively to arbitrary graph-based indexes. Consequently, it offers a valuable, readily integrable technique for accelerating diverse ANNS applications without modifying their core structure.

## 2 PRELIMINARIES

### 2.1 APPROXIMATE NEAREST NEIGHBOR SEARCH

Given a dataset $\mathcal{D}$ of high-dimensional vectors, the nearest neighbor search problem seeks to construct a data structure that, for any query vector $q$, returns its closest neighbor in $\mathcal{D}$ with minimal latency. However, as the dimensionality of $\mathcal{D}$ increases, exact search methods suffer from the *curse of dimensionality* and effectively degrade to linear-time scanning (Indyk & Motwani, 1998; Marimont & Shapiro, 1979; Clarkson, 1994). Consequently, the combination of large dataset sizes and high dimensionality motivates the approximate nearest neighbor search (ANNS), in which one trades the accuracy for speed. In this context, accuracy is measured by recall, defined as the fraction of queries for which the true nearest neighbor is correctly identified.

Existing approximate nearest neighbor search (ANNS) techniques are classified into four groups: tree-based (Bentley, 1975; Beckmann et al., 1990), quantization-based (Jegou et al., 2010; Gao & Long, 2024), hashing-based (Gionis et al., 1999; Liu et al., 2014), and graph-based methods (Jayaram Subramanya et al., 2019; Malkov & Yashunin, 2018; Harwood & Drummond, 2016). Multiple researches experimentally report that graph-based approaches outperform others by achieving higher recall and lower latency (Wang et al., 2021; Malkov & Yashunin, 2018; Fu et al., 2019).

### 2.2 GRAPH-BASED APPROXIMATE NEAREST NEIGHBOR SEARCH

Graph-based ANNS methods (Malkov & Yashunin, 2018; Fu et al., 2019; Jayaram Subramanya et al., 2019; Harwood & Drummond, 2016; Li et al., 2019) construct a *graph index* in the form of a proximity graph $G = (V, E)$ over the dataset $\mathcal{D}$, where each vector $v \in \mathcal{D}$ corresponds to a node in $V$. The edge set $E$ is determined with respect to the distance metric $\rho$ and the underlying geometric structure of $\mathcal{D}$.

Although numerous algorithms exist for constructing $G$, most share a common search rou-

---

**Algorithm 1** GREEDYBEAMSEARCH($ep, q, w, G$)

---

1: beam $\leftarrow [\,ep\,]$, visited $\leftarrow \varnothing$
2: **while** beam $\nsubseteq$ visited **do**
3:      $v \leftarrow \underset{x \in \text{beam} \setminus \text{visited}}{\arg\min} \; \rho(x, q)$
4:      visited $\leftarrow$ visited $\cup \{v\}$
5:      beam $\leftarrow$ beam $\cup \mathcal{N}(v)$
6:      **sort** beam increasingly by $\rho(\cdot, q)$
7:      beam $\leftarrow$ top-$w$ entries of beam
8: **end while**
9: **return** beam

---

tine. Given a graph index $G$ and an entry point $ep$, a simple greedy search locates a query $q$ by iteratively moving from the current node $v$ to a neighbor $x \in \mathcal{N}(v)$ that minimizes $\rho(x, q)$. The search terminates once no neighbor is closer to $q$. Graph-based methods extend this to a *greedy beam search*, as formalized in Algorithm 1. This variant maintains a beam of size $w$ tracking the $w$ closest nodes encountered, with strict greedy search recovered when $w = 1$. The search concludes when every node in the beam has already been visited.

Fast convergence of Algorithm 1 depends critically on the choice of $\mathcal{N}(v)$, how the graph index is built. A naive approach is letting $\mathcal{N}(v)$ to consist of the $M$ closest nodes of $v$. Recent works (Malkov & Yashunin, 2018; Fu et al., 2019; Jayaram Subramanya et al., 2019; Harwood & Drummond, 2016) refine this by enforcing diversification through pruning. They try to construct $\mathcal{N}(v)$ such that it consists of $M$ nodes that are not only close to $v$ but are also well separated from one another. Further details on the pruning methodology can be found on Appendix B. In practice, Algorithm 1 is invoked for each $v \in V$ to generate a candidate set, and pruning is then applied to finalize $\mathcal{N}(v)$.

## 2.3 MULTIPLE-QUERY OPTIMIZATION

For many computational tasks, processing a batch of inputs together can bring an extra dimension of optimization and higher throughput. In traditional databases, this challenge is known as the multiple-query optimization (MQO) (Sellis, 1988). MQO seeks an execution strategy that combines relational queries by exploiting shared intermediate results and eliminating redundant computation.

We illustrate MQO with a minimal example. Consider a table with attributes $a$ and $n$, denoting users' age and name, respectively, and two queries $X$ and $Y$. Query $X$ retrieves the names of individuals over 20 years old, while query $Y$ counts the number of users older than 45. Query $X$ is evaluated by applying a selection $\sigma_{a>20}$ followed by a projection $\Pi_n$, whereas query $Y$ applies $\sigma_{a>45}$ and then an aggregation to count the results. Since $\sigma_{a>20}$ yields a superset of $\sigma_{a>45}$, the intermediate result of $X$ can be reused for $Y$, reducing computation. A more detailed example is provided in Appendix C.

This example highlights the relational domain's substantial overlapping computations and ample opportunities for reuse. A rich body of work has consequently devised MQO algorithms that construct processing strategies for batches to exploit such reuse (Dokeroglu et al., 2014; Bayir et al., 2006; Trummer & Koch, 2017; Marcus et al., 2019). By contrast, the algorithmic potential of MQO in ANNS remains largely underexplored. In this paper, we bridge this gap by identifying and exploiting MQO opportunities to achieve more efficient query processing in ANNS.

## 3 ENABLING MULTIPLE-QUERY OPTIMIZATION IN ANNS

To enable *multiple–query optimization*, two key prerequisites must be satisfied:

1. **Identification of reuse opportunities.** Analyze incoming queries to uncover common intermediate computations, enabling computational reuse and efficiency.

2. **Preprocessing algorithm for query batches.** Reorder or transform queries to materialize the identified *reuse potential* and fully exploit it throughout execution.

The fundamental reason MQO is feasible in traditional databases is that the first requirement, identification of reuse opportunities, is readily satisfied. Since relations consist of tuples with fixed schemas and finite attributes, factoring each query into a relational-algebra tree reveals identical or subsumable operations that naturally emerge across queries. This structural characteristic allows overlapped computations to be identified easily through syntactic or semantic analysis of the queries.

By contrast, in approximate nearest neighbor search, a query is simply a vector of real numbers with no additional structure. Consequently, potential overlaps across distinct queries remain hidden, making the discovery of reusable computations nontrivial.

### 3.1 MAIN IDEA: MQO FRAMEWORK FOR ANNS

The key to revealing reuse opportunities lies in leveraging the distinctive properties and design details of modern ANNS algorithms. A crucial observation is that modern ANNS solutions predominantly rely on graph-based methods, wherein a greedy beam search (Algorithm 1) is initiated from an

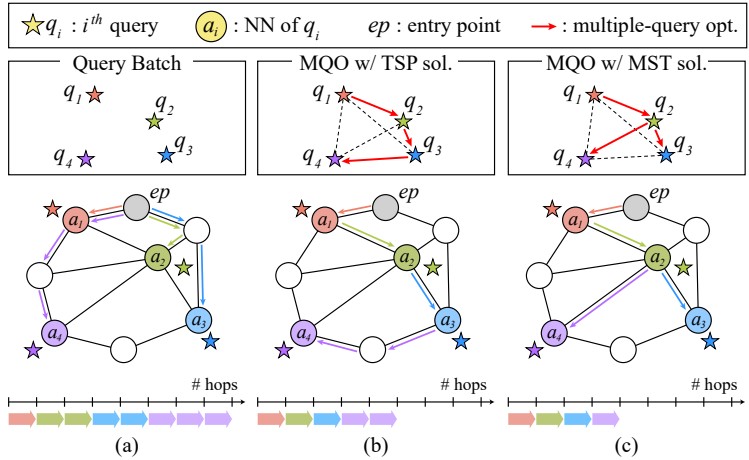

Figure 1: Overall idea for multiple-query optimization in ANNS.

entry point $ep$ determined by a predefined policy that is independent of the query. For instance, HNSW (Malkov & Yashunin, 2018) uses the topmost node in its hierarchical graph, while NSG (Fu et al., 2019) and Vamana (Jayaram Subramanya et al., 2019) use a medoid of $\mathcal{D}$ regardless of the input query. However, since proximity graph tends to connect nearby points, the choice of entry point critically influences performance. Intuitively, selecting an entry point close to the query reduces the number of hops required for search, thereby minimizing search latency.

We propose replacing the predefined entry point with one tailored to each query. While this is infeasible for an isolated query, it becomes possible when processing a batch. Consider two successive queries $q_1, q_2 \in Q$ with $\rho(q_1, q_2)$ small. Searching for $q_1$ returns its nearest neighbor $a_1$ such that $\rho(a_1, q_1)$ is small. Given that $\rho$ is a proper metric,

$$\rho(a_1, q_2) \leq \rho(a_1, q_1) + \rho(q_1, q_2),$$

guaranteeing $a_1$ to lie near $q_2$. Initializing the search for $q_2$ at $a_1$ therefore provides a high-quality entry point for $q_2$, enabling indirect reuse of computation without modifying the underlying index.

To realize and exploit such reuse potential, we must order the query batch $Q = \{q_1, \ldots, q_{|Q|}\}$ so that successive queries remain close. Formally, we seek a permutation $Q_p = (q'_1, \ldots, q'_{|Q|})$ minimizing

$$\sum_{i=2}^{|Q|} \rho(q'_{i-1}, q'_i),$$

which is precisely the open traveling salesman problem (TSP) on $Q$.

However, our mechanism does not require a single path. Once query $q_i$ yields its nearest neighbor $a_i$, that same $a_i$ can serve as the entry point for all subsequent queries. Thus it suffices to find any connected structure spanning the entire set $Q$. Such structure supports traversal from an arbitrary starting query, where the discovery sequence of oriented edges defines a processing strategy. Our objective is now to choose a set of query pairs $Q_p \subseteq \{(q, q') \mid q, q' \in Q\}$ spanning $Q$ that minimizes

$$\sum_{\{q, q'\} \in Q_p} \rho(q, q'),$$

which is exactly the classical minimum spanning tree (MST) objective. Since every hamiltonian path is a spanning tree, MST-based preprocessing is superior over TSP-based formulation. Thus, we adopt the MST approach as our final strategy. We additionally include experimental comparisons with alternative query planning strategies, with results presented in Appendix D.

Figure 1 summarizes our discussion. In (a), the standard ANNS search always begins from the predefined entry point $ep$, irrespective of the query. In (b), we apply MQO via TSP: solving the TSP on $Q$ yields a path $Q_p = \{q_1, q_2, q_3, q_4\}$, so that search for $q_1$ starts from $ep$ and each subsequent $q_i$

initializes at the nearest neighbor $a_{i-1}$ of the preceding query. In (c), we discard path constraint and utilize a tree structure: solving the MST on $Q$ gives the edge set $Q_p = \{(q_1, q_2), (q_2, q_3), (q_2, q_4)\}$, allowing queries to use the best available entry point. For instance, the search for $q_4$ starts from $a_2$ instead of $a_3$. This improved approach minimizes inter-query distances and, consequently, the total number of hops and node visits across the batch, ultimately reducing the overall latency.

## 3.2 THEORETICAL GUARANTEE

At a high level, our MQO framework posits that the distance $L$ between the entry point and a query node directly dictates the search latency. This dependence was intuitively assumed in our earlier discussion on Section 3.1 without formal justification. Theorem 1 supplies the missing link. We prove that the *expected* running time grows at least linearly with $L$, so any preprocessing that systematically shortens $L$ is guaranteed to reduce the search cost.

**Setup.** Let the dataset $\mathcal{D} = \{v_i\}_{i=1}^N$ be i.i.d. draws from a distribution on $\mathbb{R}^d$ with density $f$ supported on a region $\mathcal{W} \subset \mathbb{R}^d$ with nonempty interior. Assume $f$ admits a strictly positive lower bound $f_{\min}$ on $\mathcal{W}$. Construct a proximity graph $G$ by connecting each vector to its $M$ nearest neighbors. For simplicity, assume a strict greedy search ($w = 1$) and the Euclidean metric $\rho(x, y) = \|x - y\|_2$. Further assume a dense regime where $\ln N \gg \ln d$ and $N \gg M$.

**Theorem 1** (Average running time, lower bound). *Let $ep, q \in \mathcal{D}$ be the entry point and the query vector, respectively, and let $L = \rho(ep, q)$ be the distance between them. Suppose the greedy search process starting from $ep$ converges to $q$. Then there exists $c = c(d, f_{min}) > 0$ independent of $L$ such that the expected running time $\mathbb{E}[\mathcal{T}(ep, q)]$ satisfies*

$$\mathbb{E}[\mathcal{T}(ep, q)] \geq \left[ c\, M \left( \frac{N}{M \ln N} \right)^{\frac{1}{d}} \right] L \qquad \left( i.e., \ \ \mathbb{E}[\mathcal{T}(ep, q)] = \Omega(L) \right)$$

The main strategy of the proof is to bound the expected reduction in distance achieved by a single greedy hop. We choose a radius $R$ large enough that, with high probability, all $M$ nearest neighbors of the current node lie within the ball of radius $R$. The core insight is that, for this typical case, the greedy step decreases the distance by at most $R$ while on the rare complementary event that some neighbor lies outside this ball, the step may cover a larger gap, but no more than the total distance $L$. By balancing these two regimes, we obtain a tight upper bound on the expected progress per hop. Summing this bound over the entire path to the query then yields a matching lower bound on the expected number of hops and on the overall search complexity. The full proof is given in Appendix A.

While Theorem 1 establishes a direct link between the distance $L$ and the search cost, it does not quantify how much $L$ can be reduced by our MST-based preprocessing. Theoretical studies on Euclidean MSTs show that, for $|Q|$ points sampled uniformly from the unit hypercube, the total MST weight scales as $\mathcal{O}(|Q|^{(d-1)/d})$ (Steele & Snyder, 1989). Hence, by ordering the queries according to the MST, our MQO framework guarantees that the average inter-query distance shrinks at rate $\mathcal{O}(|Q|^{-1/d})$, leading to progressively larger per-query latency reductions as $|Q|$ grows.

## 3.3 RAPID QUERY PREPROCESSING

For batched query scenarios in which queries accumulate incrementally over a time window or are known *a priori*, preprocessing can be performed during the accumulation period. By contrast, in scenarios where the full batch is only revealed at runtime, one must minimize the end-to-end runtime, including any preprocessing overhead. In such cases, constructing the MST is prohibitively expensive, as computing all pairwise distances incurs $\mathcal{O}(|Q|^2)$ time. Since the subsequent search scales linearly with $|Q|$, the quadratic term soon dominates, precluding the use of our MQO framework.

Breaking the quadratic barrier starts from noting that constructing a *single* spanning tree is unnecessary; instead, a *spanning forest* suffices. Beginning from the entry point governed by the predefined policy, each tree root can be entered exactly once, after which the traversal proceeds within the tree according to the method proposed in Section 3.1. Thus, with $K$ trees, the predefined entry point is utilized $K$ times rather than only once.

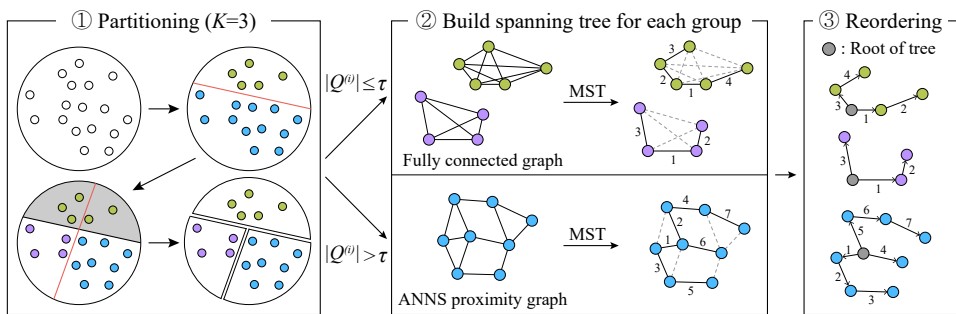

Figure 2: Overview of rapid query preprocessing scheme.

The overall procedure of our rapid preprocessing method is illustrated in Figure 2. ① We partition $Q$ into $K$ groups, $\{Q^{(1)} \ldots Q^{(K)}\}$, via a recursive random-hyperplane strategy. At each iteration, we sample a random vector in $\mathbb{R}^d$ as the normal vector of the hyperplane. Then we project the queries in the largest group onto the vector, and partition that group into two subsets according to the sign of projection. The time required to build the spanning forest, where each tree is the MST of its corresponding group, is proportional to $\sum_{i=1}^{K} |Q^{(i)}|^2$. By always bisecting the largest group, we minimize $\max_i |Q^{(i)}|$, thereby effectively reducing the overall preprocessing cost.

After partitioning, ② we build a spanning tree *within* each group. We utilize an auxiliary ANNS to further tackle the quadratic dependency. If the group size $|Q^{(i)}|$ is sufficiently small, computing the exact MST for the group of queries does not incur intolerable overhead. On the other hand, if it is above a certain threshold $\tau$, we first build a lightweight ANNS proximity graph over the group, symmetrize it to form an undirected graph, and then compute the MST on top of the ANNS graph. This confines the MST computation to a much smaller edge set, alleviating quadratic complexity.

Another important aspect worth noting is the ③ reordering step. We apply Prim's algorithm (Prim, 1957) to each group to compute its MST, which produces oriented edges in discovery order. This orientation allows us to use the nearest neighbor result of the parent query as the entry point for the child query. Although one could process queries directly in Prim's sequence, we instead reorder the forest to maximize locality for better cache utilization. For each tree, we select a root at random and perform a depth-first traversal, recording edges in the order they are first encountered. This *flattening* procedure ensures that consecutive edges correspond to spatially adjacent regions in the tree. While the total edge distance remains unchanged, the reordering mitigates cache inefficiencies caused by Prim's arbitrary discovery order, thereby improving locality and reducing runtime.

### 3.4 OVERALL ALGORITHM

---

**Algorithm 2** MQO-ANNS$(Q, G, ep, w, K, \tau)$

1: $A \leftarrow \text{EMPTYLIST}(|Q|)$
2: **if** *needRapidPreprocessing* **then**
3:     $F \leftarrow \text{MAKEFOREST}(Q, K, \tau)$
4: **else**
5:     $F \leftarrow \{\text{MST}(Q)\}$
6: **end if**
7: $Q_p \leftarrow \text{REORDER}(F)$
8: **for each** $(q_p, q_c)$ **in** $Q_p$ **do**
9:     $s \leftarrow \begin{cases} ep & \text{if } q_c \text{ is root,} \\ A[p] & \text{otherwise} \end{cases}$
10:     $A[c] \leftarrow \text{GREEDYBEAMSEARCH}(s, q_c, w, G)$
11: **end for**
12: **return** $A$

---

**Algorithm 3** REORDER$(F)$

**Require:** forest $F = \{T_1, \ldots, T_K\}$
**Ensure:** list $Q_p$ of (parent, child) pairs in DFS order with parent$= \varnothing$ for each root
1: $Q_p \leftarrow []$
2: **for each** tree $T$ **in** $F$ **do**
3:     $q_{\text{root}} \leftarrow \text{RANDSELECT}(T)$
4:     $\text{APPEND}(Q_p, (\varnothing, q_{\text{root}}))$
5:     $E \leftarrow \text{DFS}(T, q_{\text{root}})$
6:     **for each** edge $(q_{\text{parent}}, q_{\text{child}})$ **in** $E$ **do**
7:         $\text{APPEND}(Q_p, (q_{\text{parent}}, q_{\text{child}}))$
8:     **end for**
9: **end for**
10: **return** $Q_p$

---

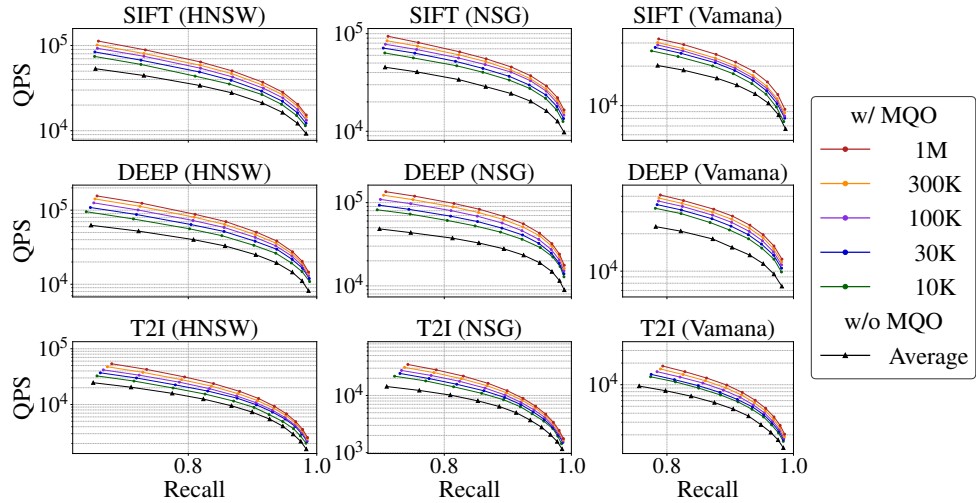

Figure 3: QPS - recall curve with varying query batch size. The reported results exclude preprocessing overhead, and MQO denotes standard MQO here.

For completeness, we outline the full framework in Algorithm 2. When end-to-end speedup is essential and rapid preprocessing is required, we build a spanning forest via partitioning and auxiliary ANNS (Line 3) as described in Section 3.3. Otherwise, computing the exact MST captures reuse opportunities to the fullest extent (Line 5). We then invoke Algorithm 3 to produce sequence of oriented query pairs $Q_p$, which defines the search plan (Line 7). Finally, we execute the search for each query (Lines 8–11). When a query is a root, we begin from the predefined entry point $ep$, and otherwise we initialize from the nearest neighbor result obtained for its parent in the tree. More detailed and complete versions of these algorithms are provided in Appendix E.

## 4 EVALUATION

We evaluate our MQO framework on three widely used graph-based ANNS indexes: HNSW (Malkov & Yashunin, 2018), NSG (Fu et al., 2019), and Vamana (Jayaram Subramanya et al., 2019). For each index, we adopt the official implementations[1], extending only the search API to allow specification of an external entry node. Unless stated otherwise, we use the default parameters when constructing each graph index. We used search on each graph index without applying MQO as the baseline. Experiments are conducted on three datasets: SIFT (uint8, 128-dim, Euclidean) (Amsaleg & Jégou, 2010), DEEP (float, 96-dim, Euclidean) (Yandex & Lempitsky, 2016; Baranchuk & Babenko, 2021a), and Yandex Text-to-Image (T2I) (float, 200-dim, inner-product) (Baranchuk & Babenko, 2021b). From the original dataset of 1B vectors, we randomly sample 1M vectors to form the dataset $\mathcal{D}$ and another 1M for the query batch $Q$, drawn from the same distribution without overlap. We report recall@1 for all experiments. The detailed setup for the evaluation can can be found on Appendix F.

### 4.1 FEASIBILITY AND EFFECTIVENESS OF THE PROPOSED MQO FRAMEWORK

We begin by evaluating our MQO framework under the assumption that the MST, which captures the full extent of reuse opportunities among queries, is provided in advance. We refer to this setting as standard MQO. While Sections 3.1 and 3.2 provide intuitive explanations and theoretical guarantees, it remains essential to verify the effectiveness of standard MQO under practical conditions—namely, on pruned proximity graphs (Section 2.2), non-Euclidean metrics, and general beam sizes ($w > 1$).

---

[1]HNSW (Malkov & Yashunin, 2018): https://github.com/nmslib/hnswlib
NSG (Fu et al., 2019): https://github.com/ZJULearning/nsg
Vamana (Jayaram Subramanya et al., 2019): https://github.com/microsoft/DiskANN

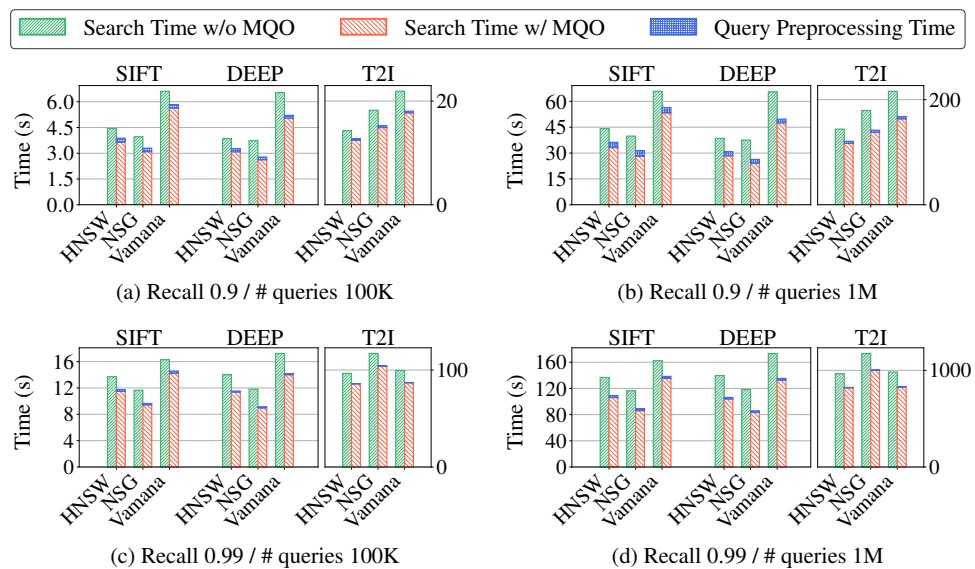

Figure 4: End-to-end time comparison for different recall and # queries. The results include preprocessing overhead, and MQO denotes lightweight MQO here.

Figure 3 presents the evaluation results, with each row corresponding to a dataset and each column to an ANNS index. In all cases, standard MQO yields consistent performance improvements, irrespective of the underlying index construction algorithm, the dataset, or the target recall level.

In particular, even at a small batch size of $|Q| = 10K$, standard MQO achieves a $1.32\times$ geometric mean speedup at a recall 0.9. Moreover, as predicted in Section 3.2, QPS further increases with batch size. For large batch of $|Q| = 1M$, the speedup rises to $1.84\times$ relative to the baseline without MQO.

We also observe a decrease in speedup as the recall target increases, due to larger beam sizes $(w)$. For instance, with $|Q| = 1M$, the speedup is $1.84\times$ at recall 0.9 but decreases to $1.62\times$ at recall 0.99. This behavior arises because the search algorithm persists until the beam is fully populated and convergence is reached. When the beams are large, the algorithm may continue exploring even after the true nearest neighbor has been discovered. Consequently, the advantage conferred by standard MQO 's improved entry points is partially diminished by the convergence-based termination criterion.

## 4.2 END-TO-END SPEEDUP

Next, we consider a scenario in which the entire query batch is revealed immediately before the search begins, leaving no opportunity for extensive preprocessing. In such cases, constructing the MST as a preprocessing step can become more expensive than the search itself, especially as the batch size increases, as illustrated in Appendix G. To address this, we bypass MST construction and instead adopt the rapid query preprocessing method introduced in Section 3.3, aiming to optimize the overall runtime across both the preprocessing and search phases. We denote this variant lightweight MQO, and compare its end-to-end performance against a baseline search without MQO.

Figure 4 presents the results. lightweight MQO achieves a $1.22\times$ geometric mean speedup over the non-MQO baseline, averaged across recall targets $\{0.9, 0.99\}$ and batch sizes $\{100K, 1M\}$. Even when including preprocessing latency, significant gains arise from reordering query execution and reassigning entry points. Notably, a net end-to-end speedup is maintained at large batch sizes (e.g., $|Q| = 1M$), as our rapid query preprocessing scales efficiently with batch size.

Specifically, when comparing speedups at recall targets of 0.9 and 0.99, we observe comparable performance. As detailed in Section 4.1, the absolute gap between MQO and the non-MQO baseline narrows at higher recall levels. However, achieving higher recall requires longer search durations, which reduces the relative impact of preprocessing. These opposing effects effectively cancel out,

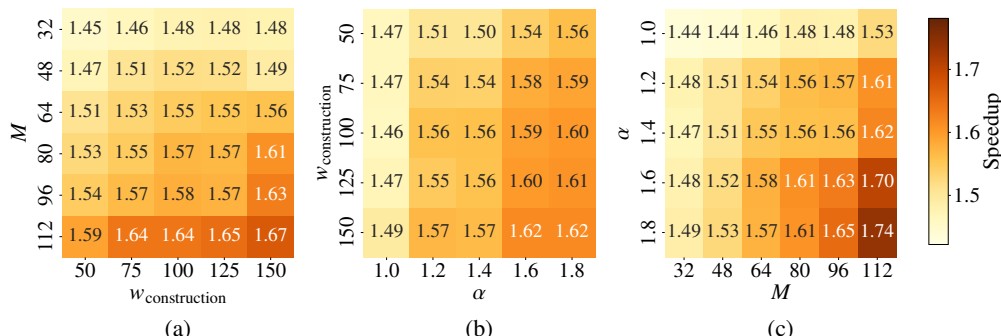

Figure 5: Geometric mean of speedup by standard MQO with Vamana for pairs of parameters (a) $w_{\text{construction}}$-$M$, (b) $\alpha$-$w_{\text{construction}}$, and (c) $M$-$\alpha$.

yielding a near-constant end-to-end speedup across the evaluated recall range. The detailed analysis of our approximation method is given in Appendix G.

### 4.3 EFFECT OF GRAPH PARAMETERS

Another question of practical interest is how the performance of our MQO framework evolves as the graph index is tuned. To investigate this, we use the DEEP dataset with $|Q| = 100$K and systematically vary the three construction knobs of Vamana—maximum out-degree $M$, the construction beam size $w_{\text{construction}}$, and pruning coefficient $\alpha$ (denoted $R$, $L$, and $\alpha$ respectively in (Jayaram Subramanya et al., 2019)). Figure 5 reports the speedup at recall 0.9 for each pairwise combination of these parameters, with the third parameter aggregated via its geometric mean.

In summary, more complex, higher-quality graphs consistently yield greater speedups. Specifically, increasing $M$ and $w_{\text{construction}}$ enhances graph size and fidelity, while raising $\alpha$ enforces more aggressive pruning. Each of these adjustments improves graph quality, thereby reducing the beam size required to achieve a given recall and, consequently, translating into higher speedup (Section 4.1). Comprehensive results on correlation between each parameter and required beam size or the beam size and the speedup are provided in Appendix H.

## 5 CONCLUSION AND DISCUSSION

In this paper, we have presented a multiple-query optimization framework toward approximate nearest neighbor search, thereby broadening the scope of optimization. By strategically reordering query execution and selecting tailored entry points for each query, our approach enhances search performance without any modification to the underlying index structure or the search algorithm.

We note that our MQO framework can achieve greater benefits in real-world scenarios. For our empirical evaluation, we assumed that both the dataset $\mathcal{D}$ and the query batch $Q$ follow the exact same distribution, which is arguably the least favorable setting for MQO, as queries exhibit no inherent locality, interleave with data points, and preclude any natural clustering. In contrast, real-world workloads often yield batched queries with strong spatial locality. A prime example is retrieval-augmented generation (RAG) systems, in which a user's original query is rewritten or augmented by multiple methods (Mao et al., 2021; Gao et al., 2023; Wang et al., 2024; Shen et al., 2024; Chan et al., 2024; Wang et al., 2025), producing a batch of related queries on which ANNS is performed. Since these queries share a common origin, they tend to cluster in the embedding space, increasing opportunities for computation reuse and amplifying the advantages of MQO.

There remains ample opportunity to extend our preliminary MQO framework. One interesting direction is to consider index construction. In graph-based ANNS, building the graph index essentially amounts to executing the search algorithm over an entire batch of queries, a setting that aligns naturally with our framework. Exploring whether the proposed framework can accelerate index building while preserving quality constitutes an intriguing avenue for future work.

## 6 REPRODUCIBILITY STATEMENT

We provide code in the supplementary material to demonstrate the methods proposed in this paper. A small dataset and a README file are included to facilitate usage. By following the provided instructions, one can reproduce the results obtained with our framework and compare them against the baseline.

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

## A  PROOF OF THEOREM 1

*Proof.*  **1. The $d$-dimensional ball.**  Define $\mathcal{B}_x^d(r)$ as the ball of radius $r$ centered at $x \in \mathbb{R}^d$:

$$\mathcal{B}_x^d(r) \;=\; \{\, y \in \mathbb{R}^d \mid \rho(x,y) \leq r \,\}.$$

Under the Euclidean metric, its volume is

$$V_d(r) \;=\; \frac{\pi^{d/2}}{\Gamma(d/2 + 1)}\, r^d \;=\; V_d(1)\, r^d.$$

Since $f(x) \geq f_{\min} > 0$ on $\mathcal{W}$, whenever $\mathcal{B}_x^d(r) \subseteq \mathcal{W}$ we have

$$\Pr[v \in \mathcal{B}_x^d(r)] \;\geq\; f_{\min}\, V_d(r) \;=\; \mu\, r^d, \qquad \mu \;:=\; f_{\min}\, V_d(1).$$

**2. Effective radius $R_\delta$.**  For any $\delta \in (0,1)$, define $R_\delta$ so that whenever $r > R_\delta$, the probability of having at least $M$ dataset points in $\mathcal{B}_x^d(r)$ is at least $1 - \delta$. Concretely, letting

$$p \;=\; \Pr\big[\, |\{\, v \in \mathcal{D} \mid v \in \mathcal{B}_x^d(r) \subseteq \mathcal{W}\}| \geq M \,\big],$$

we require $p > 1 - \delta$ given $r > R_\delta$. We show that

$$R_\delta \;=\; \left( \frac{1}{\mu\,(N-M+1)}\, \ln \frac{N^M}{\delta} \right)^{\frac{1}{d}}.$$

Write $X := |\{\, v \in \mathcal{D} : v \in \mathcal{B}_x^d(r)\}| \sim \mathrm{Bin}(N, p_0)$ with $p_0 = \Pr[v \in \mathcal{B}_x^d(r)] \geq \mu r^d$. By $(1-x) \leq e^{-x}$, we have

$$
\begin{aligned}
1 - p = \sum_{i=0}^{M-1} \binom{N}{i} p_0^i\, (1-p_0)^{N-i} &\leq \sum_{i=0}^{M-1} \binom{N}{i} (1 - \mu\, r^d)^{N-i} \\
&\leq M \binom{N}{M} (1 - \mu\, r^d)^{N-M+1} \\
&\leq M \binom{N}{M} e^{-\mu\, r^d\,(N-M+1)} \;\leq\; N^M\, e^{-\mu\, r^d\,(N-M+1)}.
\end{aligned}
$$

Then, from

$$r > R_\delta \;\implies\; \mu\, r^d\,(N - M + 1) \;>\; \ln \frac{N^M}{\delta} \;\implies\; N^M\, e^{-\mu\, r^d\,(N-M+1)} \;<\; \delta,$$

we conclude $r > R_\delta$ implies $1 - p < \delta$, so indeed $p > 1 - \delta$.

**3. Expected decrease in distance.**  Consider a single hop (iteration) of the greedy search process. Let

$$\Delta \;=\; \rho(x,\, q) \;-\; \rho(\mathrm{next}(x),\, q).$$

be the decrease in distance after we travel from the current node $x$ to one of its neighbor $\mathrm{next}(x)$ that is closest to the query.

For a given $\delta$, with probability higher or equal than $1 - \delta$, all the M nearest neighbors reside inside $\mathcal{B}_x^d(R_\delta)$. In such cases, the distance to the query cannot decrease more than $R_\delta$ in that hop. Otherwise (an event of probability at most $\delta$), some points among the $M$ nearest neighbors reside outside $\mathcal{B}_x^d(R_\delta)$. For such cases, the decrease in distance can be bounded above trivially by $L$. Thus, for any $\delta \in (0,1)$ we have :

$$\mathbb{E}\big[\Delta\big] \;=\; \mathbb{E}\Big[\rho\big(x,q\big) - \rho\big(\mathrm{next}(x),\, q\big)\Big] \;\leq\; R_\delta \;+\; \delta\, L.$$

To obtain a tight upper bound on $\mathbb{E}[\Delta]$, we choose

$$\delta^* \;=\; \frac{1}{L}\left( \frac{M \ln N}{\mu\, N} \right)^{\frac{1}{d}},$$

so that

$$\delta^* L = \left( \frac{M \ln N}{\mu N} \right)^{\frac{1}{d}}.$$

Recall that

$$R_\delta = \left( \frac{1}{\mu(N - M + 1)} \ln \frac{N^M}{\delta} \right)^{\frac{1}{d}}.$$

Substituting $\delta = \delta^*$ yields

$$R_{\delta^*} = \left( \frac{1}{\mu(N - M + 1)} \left\{ M \ln N + \ln L + \frac{1}{d} \ln(\mu N) - \frac{1}{d} \ln(M \ln N) \right\} \right)^{\frac{1}{d}}.$$

As $\ln N \gg \ln d$, $\ln L$ is negligible compared to $\ln N$. Thus the terms inside the logarithm can be bounded above by $(M + 1) \ln N$ when $N$ is large. Furthermore, as $N \gg M$, we have $\frac{M+1}{N-M+1} \le \frac{2M}{N}$. Thus,

$$R_{\delta^*} \le 2^{\frac{1}{d}} \left( \frac{M \ln N}{\mu N} \right)^{\frac{1}{d}}$$

Choosing $\delta = \delta^*$ thus balances the contribution of two terms $R_{\delta^*}$ and $\delta^* L$ so that neither dominates.

Thus, the expected decrease in distance per hop satisfies

$$\mathbb{E}[\Delta] \le R_{\delta^*} + \delta^* L \le (1 + 2^{1/d}) \left( \frac{M \ln N}{\mu N} \right)^{\frac{1}{d}}$$

This completes the derivation of the upper bound on the expected decrease in distance per hop.

**4. Expected running time of the algorithm.** Let $\tau$ be the sequence of nodes visited in the greedy search algorithm.

$$\tau = \left( x_0 = ep,\ x_1,\ x_2,\ \ldots,\ x_{T-1},\ x_T = q \right),$$

where $T$ is the stopping time, the number of hops needed for the algorithm to reach $q$.

$$T = \min\{\, i \ge 0 : x_i = q \,\}.$$

Also let

$$\Delta_i = \rho(x_{i-1}, q) - \rho(x_i, q)$$

denote the decrease in distance from $q$ at the $i$-th hop.

As we prove for the case where greedy search successfully converged to $q$, by construction $\Delta_i \ge 0$ and $\sum_{i=1}^{T} \Delta_i = L$.

In the previous step, we showed that for every $x$

$$\mathbb{E}\big[\Delta_i \,\big|\, x_{i-1} = x\big] \le \overline{\Delta}, \qquad \overline{\Delta} := \left(1 + 2^{1/d}\right) \left( \frac{M \ln N}{\mu N} \right)^{1/d}.$$

Consequently

$$\mathbb{E}[\Delta_i] \le \overline{\Delta} \quad \text{for all } i \ge 1.$$

With the non-negativity of the $\Delta_i$, Tonelli's theorem gives us

$$L = \mathbb{E}\Big[\sum_{i=1}^{T} \Delta_i\Big] = \sum_{i=1}^{\infty} \mathbb{E}\big[\Delta_i \mathbf{1}_{\{i \le T\}}\big] \le \sum_{i=1}^{\infty} \mathbb{E}[\Delta_i] \le \overline{\Delta} \sum_{i=1}^{\infty} \Pr\{T \ge i\} = \overline{\Delta}\, \mathbb{E}[T].$$

Therefore,

$$\mathbb{E}[T] \ge \frac{L}{\overline{\Delta}} = \frac{\mu^{1/d}}{1 + 2^{1/d}} \left( \frac{N}{M \ln N} \right)^{1/d} L.$$

Connecting the runtime with the number of hops is straightforward. Since at each hop the algorithm inspects $M$ edges and a single $\ell_2$ distance evaluation in $\mathbb{R}^d$ costs $\Theta(d)$ arithmetic operations, there exists an absolute constant $\alpha > 0$ such that the expected running time satisfies

$$\mathbb{E}[\mathcal{T}(ep, q)] \ge (\alpha\, d) M\, \mathbb{E}[T].$$

Combining with the bound on $\mathbb{E}[T]$ and absorbing $\alpha$ into the constant gives

$$\mathbb{E}[\mathcal{T}(ep, q)] \geq \underbrace{\frac{\alpha\, d\, \mu^{1/d}}{1 + 2^{1/d}}}_{=:\ c(d, f_{\min})}\, M \left(\frac{N}{M \ln N}\right)^{1/d} L\ .$$

This matches the statement of Theorem 1.

$\square$

## B  PRUNING IN GRAPH-BASED ANNS

Instead of using a naive $k$NN graph as a graph index, ANNS methods employ an additional pruning step. The pruning step is inspired by the relative neighborhood graph (RNG) (Toussaint, 1980) and its relaxed variant, the sparse neighborhood graph (SNG) (Arya & Mount, 1993). In an RNG, an edge $(v_i, v_j)$ is considered redundant and pruned if there exists a third node $v$ such that $(v_i, v_j)$ is the longest side of triangle $\triangle v_i v_j v$. Equivalently, an edge $(v_i, v_j)$ is retained in RNG if

$$\forall\, v \in V,\ \rho(v, v_i) \geq \rho(v_i, v_j) \text{ or } \rho(v, v_j) \geq \rho(v_i, v_j).$$

The SNG applies the similar principle with fewer eliminations so that it guarantees the strict greedy search, GREEDYBEAMSEARCH($ef, q, 1, G$), always converge to its true nearest neighbor for all $q \in \mathcal{D}$. This desirable property motivates the use of an SNG as the underlying proximity graph for many state-of-the-art ANNS methods (Malkov & Yashunin, 2018; Fu et al., 2019; Jayaram Subramanya et al., 2019; Harwood & Drummond, 2016).

Constructing the exact SNG is computationally prohibitive, as it requires applying the pruning rule to the full node set $V$ for every $v \in V$. Instead, ANNS methods approximate SNG by invoking Algorithm 1 during graph construction to obtain a compact set of candidates per node and running pruning only on this smaller set. For instance, (Malkov & Yashunin, 2018) prune edges within the beam returned by Algorithm 1, whereas (Fu et al., 2019) and (Jayaram Subramanya et al., 2019) apply pruning to the accumulated visited set.

## C  MQO ON TRADITIONAL DATABASES

Here, we show a further demonstration of *multiple-query optimization* upon relational databases with an extended example from Section 2.3. Table `User` stores a unique identifier *uid*, each customer's *name* and their *age*. `SalesOrder` records individual orders with a primary key *oid*, a foreign-key *uid* that links back to `User`, and a Boolean *valid* field that is `true` when the order has not been canceled.

| Listing 1: Query A | Listing 2: Query B |
|---|---|
| **SELECT** U.name | **SELECT** **COUNT**($*$) |
| **FROM** User U, SalesOrder S | **FROM** User U, SalesOrder S |
| **WHERE** U.uid = S.uid | **WHERE** U.uid = S.uid |
| **AND** U.age > 20 | **AND** U.age > 45 |
| **AND** S.valid = **TRUE** | **AND** S.valid = **TRUE** |

The queries can be written down in relational algebra as below.

$$A = \Pi_{name}\left(\sigma_{age>20}\left(U\right) \bowtie \sigma_{valid=\text{true}}\left(S\right)\right)$$

$$B = \gamma_{\text{COUNT}}\left(\sigma_{age>45\,\wedge\,valid=\text{true}}\left(U \bowtie S\right)\right) \tag{1}$$

$$= \gamma_{\text{COUNT}}\left(\sigma_{age>45}\left(U\right) \bowtie \sigma_{valid=\text{true}}\left(S\right)\right) \tag{2}$$

Both queries rely on the same base join $J = U \bowtie \sigma_{valid=\text{true}}(S)$. Query B applies the stricter predicate $age > 45$, which implies $age > 20$; hence its input is a *subset* of Query A's. Computing $J$ once and sharing it therefore avoids a second full scan of `SalesOrder` and a second join.

Eq. (1) pushes both predicates into the join, while Eq. (2) postpones the age test on `User`. If users older than 45 are rare, Eq. (1) produces fewer rows and is usually cheaper to share; if filtering by age is inexpensive but reading `SalesOrder` dominates, Eq. (2) may be preferable. Cost estimates stored in catalog statistics guide the optimizer toward the better choice.

Early studies framed MQO as a shortest-path search over plan graphs (Sellis, 1988), followed by meta-heuristic techniques such as genetic algorithms (Bayir et al., 2006) and exact formulations using integer linear programming (Dokeroglu et al., 2014). Modern work augments these rules with learning-based models that predict when sharing pays off under shifting workloads. Regardless of method, the key steps are (i) detect overlapping sub-trees like $J$ and (ii) select the reuse strategy that minimizes total runtime and I/O.

## D COMPARISON OF QUERY PLANNING STRATEGIES

We compare our MST-based query planning (MST) with other possible solutions, evaluating the impact of different preprocessing methods on query search performance.

As baselines, we considered not applying MQO, k-means clustering-based planning (Clustering) and a traveling salesman problem-based planning (TSP). Since solving exact TSP is infeasible, we implemented it based on the Adaptive Large Neighborhood Search algorithm.

We used batch size of 10K at recall@1 = 0.9, and the results are shown in Table 1. To ensure a fair comparison, the preprocessing time across all strategies was unified by setting it equal to the time required for MST construction. The reported results do not include preprocessing overhead.

Table 1: Search latency (s) when applying different query planning strategies.

| Dataset | Index | w/o MQO | Cluster | TSP | MST |
|---------|-------|---------|---------|-----|-----|
| DEEP | HNSW | 0.402 | 0.338 | 0.315 | **0.298** |
| | NSG | 0.431 | 0.302 | 0.287 | **0.276** |
| | Vamana | 0.621 | 0.516 | 0.477 | **0.467** |
| SIFT | HNSW | 0.477 | 0.424 | 0.395 | **0.376** |
| | NSG | 0.402 | 0.333 | 0.306 | **0.299** |
| | Vamana | 0.707 | 0.667 | 0.589 | **0.571** |
| T2I | HNSW | 1.369 | 1.214 | 1.145 | **1.126** |
| | NSG | 2.012 | 1.678 | 1.592 | **1.573** |
| | Vamana | 2.206 | 1.834 | **1.725** | 1.735 |

As expected, the MST-based planning achieves the best performance overall as it minimizes the distance between entry point and the query compared to other methods.

# E    OVERALL ALGORITHM

---

**Algorithm 4** MQO-ANNS$(Q, G, ep, w, K, \tau)$

---

**Require:** Query batch $Q = \{q_1, \ldots, q_{|Q|}\}$, graph index $G$, predefined entry point $ep$, beam size $w$, desired number of groups $K$, size threshold $\tau$
**Ensure:** array $A[1 \ldots |Q|]$ with the search result for every query
1:  $A \leftarrow$ EMPTYLIST$(|Q|)$
2:  **if** *needRapidPreprocessing* **then**
3:      $F \leftarrow$ MAKEFOREST$(Q, K, \tau)$
4:  **else**
5:      $F \leftarrow \{\text{MST}(Q)\}$
6:  **end if**
7:  $Q_p \leftarrow$ REORDER$(F)$
8:  **for each** $(q_p, q_c)$ **in** $Q_p$ (left–to–right) **do**
9:      **if** $q_c$ is the root **then**
10:         $A[c] \leftarrow$ GREEDYBEAMSEARCH$(ep, q_c, w, G)$
11:     **else**
12:         $A[c] \leftarrow$ GREEDYBEAMSEARCH$(A[p], q_c, w, G)$
13:     **end if**
14: **end for**
15: **return** $A$

---

**Algorithm 5** MAKEFOREST$(Q, K, \tau)$

---

**Require:** Query batch $Q$, desired number of groups $K$, size threshold $\tau$
**Ensure:** A spanning forest $F = \{T_1, \ldots, T_K\}$ of $Q$
1:  $F \leftarrow \varnothing$
2:  $\{Q^{(1)}, \ldots, Q^{(K)}\} \leftarrow$ PARTITION$(Q, K)$
3:  **for** $i \leftarrow 1$ **to** $K$ **do**
4:      **if** $|Q^{(i)}| \leq \tau$ **then**                                         ▷ small group: use exact MST
5:          $T_i \leftarrow$ MST$(Q^{(i)})$
6:      **else**                                                                            ▷ large group: use ANNS
7:          $H_i \leftarrow$ BUILDANNSGRAPH$(Q^{(i)})$
8:          $H_i \leftarrow$ SYMMETRIZE$(H_i)$                       ▷ make the ANNS graph undirected
9:          $T_i \leftarrow$ MST$(H_i)$
10:     **end if**
11:     $F \leftarrow F \cup \{T_i\}$                                     ▷ add component tree to the forest
12: **end for**
13: **return** $F$

---

**Algorithm 6** REORDER$(F)$

---

**Require:** forest $F = \{T_1, \ldots, T_K\}$                ▷ $T_i$ denotes a tree, represented by set of edges
**Ensure:** list $Q_p$ of (parent, child) pairs in DFS order with parent$= \varnothing$ for each root
1:  $Q_p \leftarrow [\,]$
2:  **for each** tree $T$ **in** $F$ **do**
3:      $q_r \leftarrow$ CHOOSEROOT$(T)$                                        ▷ pick any node as root
4:      APPEND$(Q_p, (\varnothing, q_r))$                                      ▷ add edge for root
5:      **for each** edge $(q_p, q_c)$ **in** DFS$(T, q_r)$ **do**      ▷ DFS on T starting from $q_r$
6:          APPEND$(Q_p, (q_p, q_c))$
7:      **end for**
8:  **end for**
9:  **return** $Q_p$

---

---

**Algorithm 7** PARTITION$(Q, K)$

---

**Require:** Query batch $Q \subset \mathbb{R}^d$, desired number of groups $K$
**Ensure:** $K$ disjoint subsets $\{Q^{(1)}, \ldots, Q^{(K)}\}$ with $\bigcup_{i=1}^K Q^{(i)} = Q$
 1: $\mathcal{C} \leftarrow \{Q\}$          ▷ current collection of groups
 2: **for** $t \leftarrow 1$ **to** $K - 1$ **do**
 3:      $Q^{\max} \leftarrow \arg\max_{C \in \mathcal{C}} |C|$          ▷ pick the largest group
 4:      $\mathbf{h} \sim \text{Uniform}(\mathbb{S}^{d-1})$          ▷ (d-1)-dimensional hyperplane
 5:      $A \leftarrow \varnothing, \; B \leftarrow \varnothing$
 6:      **for all** $q \in Q^{\max}$ **do**          ▷ split by the hyperplane
 7:          **if** $\mathbf{h} \cdot q \geq 0$ **then**
 8:              $A \leftarrow A \cup \{q\}$
 9:          **else**
10:              $B \leftarrow B \cup \{q\}$
11:          **end if**
12:      **end for**
13:      $\mathcal{C} \leftarrow (\mathcal{C} \setminus \{Q^{\max}\}) \cup \{A, B\}$          ▷ replace $Q^{\max}$ with its two children
14: **end for**
15: **return** $\mathcal{C}$

---

## F  ADDITIONAL EXPERIMENTAL SETTINGS

We evaluate our MQO framework on three widely used graph-based ANNS indices: HNSW (Malkov & Yashunin, 2018), NSG (Fu et al., 2019), and Vamana (Jayaram Subramanya et al., 2019). For each index, we use the official implementation[2] without modifying its core indexing or search algorithms, extending only the search API to accept an external entry node.

NSG and Vamana are planar proximity graphs. Thus we simply integrate MQO by replacing their default entry points with those selected by our framework. HNSW, in contrast, employs a multi-layer hierarchy. The search begins at the topmost layer, performs a strict greedy search, and propagates the nearest-neighbor result downward through each successive layer until the bottom layer is reached. To support MQO in HNSW, we retain the original hierarchical search for the $K$ queries using the predefined entry point (Section 3.3), and for all remaining queries we initiate search directly at the bottom layer node determined by preceding query results.

We used the following parameters to construct each proximity graph. For HNSW (Malkov & Yashunin, 2018), we adopted the configuration reported in their paper, setting $M = 16$ and $ef_{\text{construction}} = 500$. For NSG (Fu et al., 2019) and Vamana (Jayaram Subramanya et al., 2019), we used the default parameter settings provided in their official implementations. For NSG, we first constructed the EFANNA (Fu & Cai, 2016) $k$-NN graph using the parameters $K = 200$, $L = 200$, iter $= 10$, $S = 10$, and $R = 100$. The NSG was then built with $L = 40$, $R = 50$, and $C = 500$. For Vamana, we used $R = 64$, $L = 100$ and $\alpha = 1.2$.

All experiments were conducted on an AMD Ryzen Threadripper PRO 7985WX CPU, single core and 1024GB of main memory.

---

[2]HNSW (Malkov & Yashunin, 2018): `https://github.com/nmslib/hnswlib`
NSG (Fu et al., 2019): `https://github.com/ZJULearning/nsg`
Vamana (Jayaram Subramanya et al., 2019): `https://github.com/microsoft/DiskANN`

## G  MORE ON APPROXIMATION METHOD

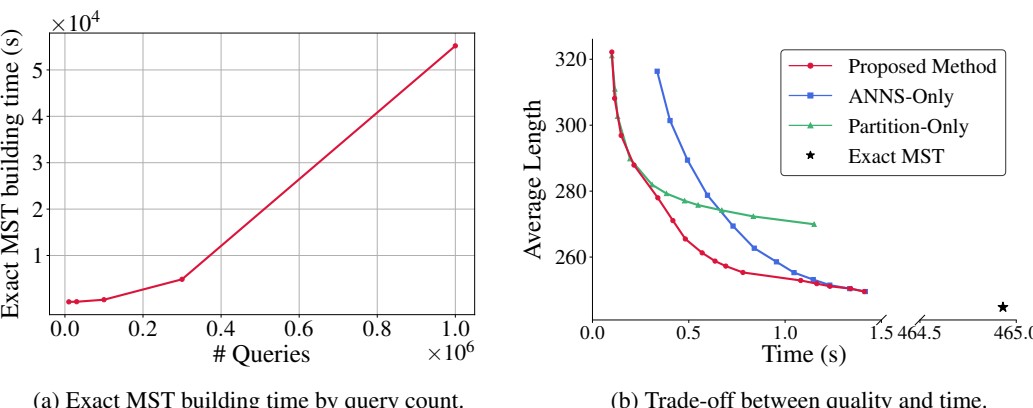

(a) Exact MST building time by query count.   (b) Trade-off between quality and time.

Figure 6: Necessity of our rapid query preprocessing method.

In this section, we illustrate the necessity and the details of our rapid preprocessing method introduced in Section 3.3. Figure 6a shows the time required to construct the exact minimum spanning tree for a query set from the SIFT dataset. While MST construction is feasible for small batch sizes, the quadratic time complexity quickly renders it impractical as the batch size grows. This observation motivates the need for a rapid preprocessing approach when limited preprocessing time is available.

Figure 6b demonstrates both the necessity and effectiveness of combining a partitioning strategy with an auxiliary ANNS proximity graph. Throughout this paper, we adopt HNSW as our auxiliary graph, since NSG construction requires a prior kNN graph and Vamana incurs multiple construction rounds. We plot the preprocessing time on the horizontal axis and the quality of the resulting forest—quantified by the average edge length—on the vertical axis. To account for the number of trees (groups), we compute a weighted average of the batch's average inter-query distance and the forest's average edge length, weighting the former by the number of trees and the latter by the batch size minus the number of trees. The curve labeled *ANNS-Only* varies ANNS parameters without partitioning, whereas *Partition-Only* varies the number of partitions without using ANNS. The curve labeled *Proposed Method* consists of Pareto-optimal points obtained by jointly employing partitioning and the auxiliary proximity graph. As the figure clearly shows, neither strategy alone suffices: integrating both yields a synergistic improvement in the quality–time trade-off.

Next, we detail the hyperparameters associated to the approximation method. At first glance, tuning the method may appear challenging due to the number of hyperparameters: the group-size threshold $\tau$, the number of groups $K$, and the auxiliary ANNS (HNSW) construction parameters. While extensive tuning can yield marginal gains, we found that fixing $\tau = 500$ and setting $M = ef_{\text{construction}} = 7$ for HNSW, while varying only $K$, is sufficient. For $K$ letting $K = |Q|/250$ is a good starting point, with further tuning yielding only slight improvements in overall speedup.

## H  MORE ON GRAPH PARAMETER EFFECT

We expand Section 4.3 with the raw measurements and the statistical tests.

Table 2: Partial association between graph parameters and residual speedup

| Parameter | $\rho$ | $p$ |
|---|---|---|
| $M$ | **0.56** | $\mathbf{7.4 \times 10^{-14}}$ |
| $w_{\text{construction}}$ | -0.18 | 0.03 |
| $\alpha$ | -0.11 | 0.20 |

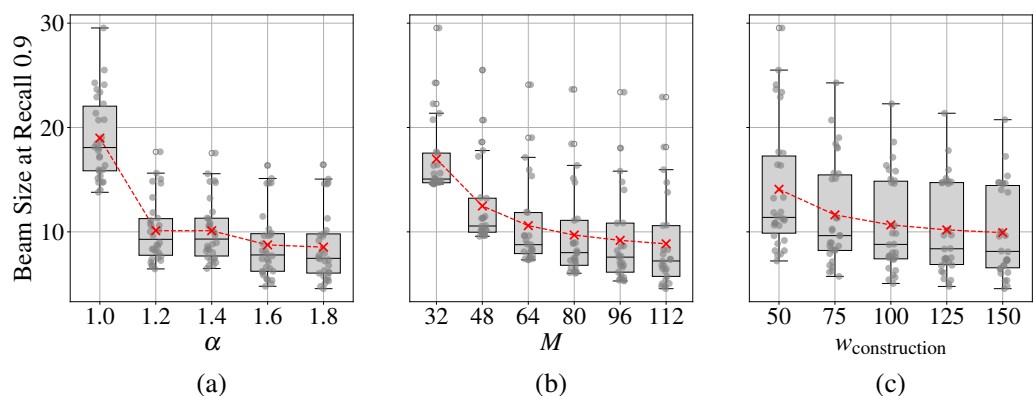

Figure 7: Required search beam versus (a) $\alpha$, (b) $M$, and (c) $w_{\text{construction}}$ on 1M DEEP dataset. Red cross marks indicate the average value.

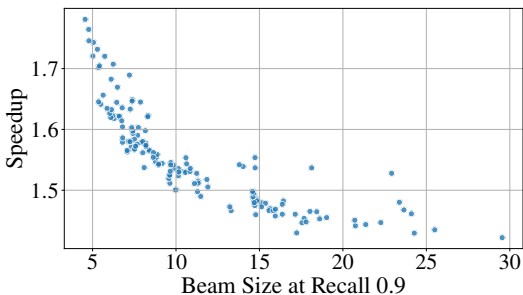

Figure 8: Speedup under varying beam size (recall 0.9).

Figure 7 shows the distribution of the beam required for 0.9 recall. All parameters continuously downscale the beam.

Figure 8 plots speedup against the required beam for every configuration. The points hug a monotonically decreasing curve, justifying to set the beam width as the primary covariate.

After regressing speedup on beam width we compute Spearman partial correlations between the residuals and each knob; results appear in Table 2. Only $M$, the out-degree limit is found to be positively correlated. $w_{\text{construction}}$ shows $p = 0.03$ which is marginally significant, but falls below the threshold with Bonferroni correction. $\alpha$ is found uninfluential other way than the search beam.

# I   PERFORMANCE FOR TOP-10 SEARCH

In this section, we examine the effect of our MQO framework when applied to top-10 search. Since the framework initializes the search from an entry point closer to the query, it enables the greedy beam search to converge more quickly, regardless of the final number of neighbors retrieved.

We evaluate search performance using the NSG index with a batch size of 100 K at recall@10=0.99, and include results for top-1 neighbor search for comparison. Preprocessing overhead is not included in the repored results (standard MQO). The results are presented in Table 3.

Although small differences are observed, the general tendency of the speedup is the same for top-1 and top-10 search.

Table 3: Search latency (s) for top-$k$ search with and without MQO (preprocessing overhead excluded).

| Top-$k$ search | Dataset | Latency w/o MQO (s) | Latency w/ MQO (s) | Speedup |
|---|---|---|---|---|
| $k = 10$ | DEEP | 14.741 | 9.101 | 1.62 |
| | SIFT | 13.578 | 9.431 | 1.44 |
| | T2I | 117.364 | 87.856 | 1.34 |
| $k = 1$ | DEEP | 14.650 | 8.998 | 1.63 |
| | SIFT | 13.397 | 9.227 | 1.45 |
| | T2I | 115.221 | 87.197 | 1.32 |

## J   MULTI-THREAD EXTENSION OF THE FRAMEWORK

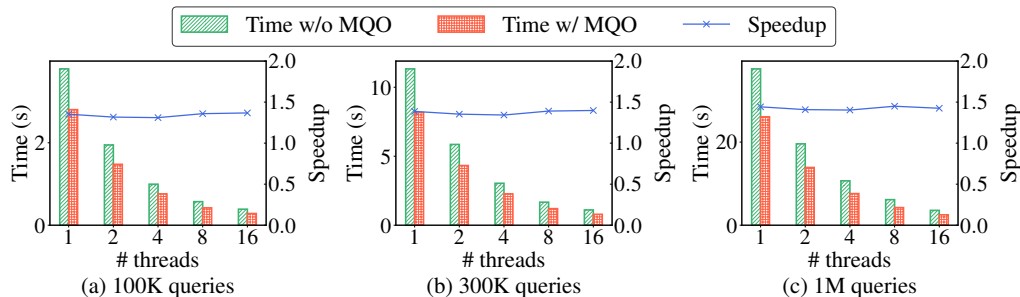

(a) 100K queries      (b) 300K queries      (c) 1M queries

Figure 9: Comparing MQO against non-MQO baselines on multi-threaded configuration. Here MQO denotes lightweight MQO and includes both the time for preprocessing and the actual search.

To demonstrate our algorithmic advancement and computational savings, the experiments presented in the main text were conducted in a single-threaded manner. Nevertheless, we emphasize that the proposed MQO framework is inherently amenable to parallelization. Since our MQO framework partitions the query batch into $K$ groups, both query preprocessing and nearest neighbor search can leverage inter-group parallelism by assigning a thread to each partitioned group.

We conducted a simple experiment to evaluate the scalability of our framework with respect to the number of threads. We compared our lightweight MQO with inter-group parallelism to a baseline that assigns each query to a separate thread without applying MQO. Parallelization was implemented using OpenMP (Dagum & Menon, 1998). The experiment was conducted on the DEEP dataset using the NSG graph index, targeting a recall of 0.9. We present the end-to-end time comparison in Figure 9. As shown, our lightweight MQO with inter-group parallelism exhibited constant speedup over the baseline across varying thread counts, demonstrating that our framework remains efficient under parallel execution.

In addition, we note that even when only small groups are used, or using $K = 1$ in the extreme case, intra-group parallelization remains feasible. In these scenarios, one can employ parallel algorithms for minimum spanning tree construction (Bentley, 1980) or ANNS proximity-graph construction (Manohar et al., 2024; Harwood & Drummond, 2016; Fu et al., 2019) during preprocessing. Furthermore, during the search phase, the branching degree of each node in the spanning tree over the query group presents additional avenues for parallel execution.

