# OpenReview forum: "Rethinking Multiple-Query Optimization for Approximate Nearest Neighbor Search"
_ICLR.cc/2026/Conference — ICLR 2026 Conference Withdrawn Submission_

### Official Review · Reviewer_ddWr · 2025-10-24

**Soundness:** 2
**Presentation:** 3
**Contribution:** 2
**Rating:** 2
**Confidence:** 5

**Summary:**

The task the authors consider is graph-based approximate nearest neighbor search (ANNS) in a batch setting with a sequential computation, i.e., they assume that the computations of the earlier queries can be used to speed up subsequent queries. The authors propose a method where a batch of queries is first partitioned by random hyperplanes, and then a forest of minimum spanning trees (MST) is obtained by building a MST within each group. The nearest neighbor of the parent of a query point in a MST is used as an entry point for a beam search in a graph. The experimental results of the article verify that on NSG and Vamana graphs, the proposed entry point selection method is more efficient than using the medoid of the data set as an entry point. In addition, when the proposed method is used the select an entry point from the bottom layer of the HNSW graph, the results show that it is more efficient than the standard HNSW graph traversal.

**Strengths:**

The article is well-written, and enough details are provided so that the method and the motivation behind it can be easily understood. The proposed method seems to be an improvement over using the medoid of the data set as an entry point.

**Weaknesses:**

Rather than presenting a novel multiple-query optimization framework for ANNS as claimed, the authors simply propose a novel seed (entry point) selection method for graph-based ANNS. Using an auxiliary index structure to select an entry point of beam search is a common technique in graph-based ANNS. A survey by Azizi et al., (2025, p. 8-9) lists 7 different seed selection methods from the literature: for instance, $k$-d trees (Munoz et al., 2019), LSH (Jin et al., 2014), and balanced $k$-means trees (Wang et al., 2012) have been applied  for seed selection. According to the experiments by Azizi et al. (2025, p. 17-18), these more advanced seed selection methods outperform using the medoid of the data set as the seed. Thus, in addition to the simple baseline where the medoid is used as the entry point, the proposed method should be compared to the stronger baselines mentioned by Azizi et al. (2025).

The main difference to the aforementioned seed selection methods is that the auxiliary index structure (MST) is both built and traversed on the query phase (before a batch of queries is executed), whereas in the earlier methods it is built in advance and only traversed at the query phase. Another important difference is that the proposed method is only applicable for batch queries, whereas the earlier seed selection methods are applicable both for single and batch queries.

In addition, the multi-threaded experimental setup of Appendix J should be a default for batch queries. Constraining batch queries to a  single-thread execution is equivalent to running all the queries of the batch sequentially, which misses the point of testing ANNS algorithms on batch queries, which is to study how efficiently the proposed method can utilize the computational resources available. For instance, the experimental setups of the GPU ANNS articles mentioned in the manuscript naturally allow multi-threaded execution for batch queries (using GPU).


References:

Azizi, Ilias, Karima Echihabi, and Themis Palpanas. "Graph-based vector search: An experimental evaluation of the state-of-the-art." Proceedings of the ACM on Management of Data 3.1 (2025): 1-31.

Jin, Zhongming, et al. "Fast and accurate hashing via iterative nearest neighbors expansion." IEEE transactions on cybernetics 44.11 (2014): 2167-2177.

Munoz, Javier Vargas, et al. "Hierarchical clustering-based graphs for large scale approximate nearest neighbor search." Pattern Recognition 96 (2019): 106970.

Wang, Jing, et al. "Scalable k-nn graph construction for visual descriptors." 2012 IEEE Conference on Computer Vision and Pattern Recognition. IEEE, 2012.

**Questions:**

I do not have any questions for the authors.

---

### Official Review · Reviewer_W38H · 2025-10-30

**Soundness:** 2
**Presentation:** 2
**Contribution:** 2
**Rating:** 2
**Confidence:** 4

**Summary:**

The authors propose an algorithmic MQO framework for graph-based ANNS, which is generally applicable to existing graph-based methods. Their approach constructs a minimum spanning tree over the query vectors and initializes each query’s search entry using the nearest neighbor returned by its parent in the tree, enabling computation reuse. Experiments are conducted to validate the effectiveness of the proposed framework.

**Strengths:**

S1: ANNS is an important topic, and any contribution to accelerating ANNS is appreciated.

S2: The authors validate the effectiveness of the proposed framework on several real datasets.

**Weaknesses:**

W1. I wonder whether the technique proposed in this paper is meaningful for ANNS. In my opinion, in a batch query scenario, multi-core processing can efficiently handle multiple queries simultaneously (see [a] for performance results in the batch query scenario). More importantly, some state-of-the-art quantization-based approaches, such as ScaNN [b], may perform very well in batch query scenarios. Currently, I do not see sufficient motivation for performing additional query preprocessing, particularly the reordering operation, for multiple queries.

W2. In the experiments, the authors only compare the proposed technique with some basic similarity graphs. It is recommended that they also compare the proposed method experimentally with quantization-based approaches, such as ScaNN [b] and RabitQ (which has been cited in the paper), as both approaches claim significant performance improvements over basic similarity graphs. Additionally, some similarity graphs [a][c] have been reported to achieve 2X–3X performance improvements on HNSW. The authors are recommended to either experimentally compare with these works or reasonably explain why these existing approaches are orthogonal to the proposed technique.

[a] Similarity search in the blink of an eye with compressed indices. VLDB 2023

[b] Accelerating large-scale inference with an isotropic vector quantization. ICML 2020

[c] Probabilistic routing for graph-based approximate nearest neighbor search. ICML 2024

**Questions:**

Please address the concerns listed in the weaknesses. Additionally, I recommend that the authors conduct experiments on GIST, GloVe, as the search difficulty of SIFT and DEEP datasets is relatively low, and T2I is not a standard dataset for ANNS evaluation.

---

### Official Review · Reviewer_3CKX · 2025-10-31

**Soundness:** 2
**Presentation:** 3
**Contribution:** 2
**Rating:** 2
**Confidence:** 4

**Summary:**

The authors propose a method for speeding up graph-based approximate nearest neighbor search in the scenario where queries arrive in batches rather than separately. The batch scenario allows for opportunities in identifying commonalities between the queries that can be exploited to share work during the graph traversal, speeding up the overall query process.

The proposed method works by building a minimum spanning tree from the queries. By ordering the queries according to the MST, previous results can be used as entry points to the graph for faster traversal in subsequent queries. To minimize end-to-end query time, the authors propose partitioning the queries into groups and building the MST for each group separately. In their experiments, the authors demonstrate that the proposed method yields an average improvement of 1.22x across batches of sizes 100K and 1M.

**Strengths:**

I like the proposed method conceptually and the method is simple to implement in practice. The method applies generally to graph-based methods and the authors demonstrate this by applying it to three different algorithms, HNSW, NSG, and Vamana. For all three methods, for large batch sizes, the method yields a non-trivial speed up. Overall, the paper is well-written and easy to read.

**Weaknesses:**

The paper does not give strong evidence on its actual practicality. End-to-end speed up is demonstrated only on batch sizes of 100K and 1M and speedups are demonstrated on lower batch sizes only when the preprocessing time is excluded. In most practical deployments where response times are critical, batch sizes are much smaller. Also some common applications for which you can imagine a large batch size, e.g. $k$-nn graph construction or clustering, have specialized methods that work much faster. For example, for the latter, see related work [1] where graph-based approximate nearest neighbor search in the assignment step of $k$-means with large $k$ is sped up by seeding the graph traversal from previous assignments.

Moreover, a natural question a reader has is also whether there is any benefit to this framework when multiple threads are used. This is answered only in the very end of the Appendix and not referred to anywhere in the text. The authors demonstrate that the speedup remains the same, but this hinges upon there being enough groups such that it's meaningful to parallelize over them. The section talks about intra-group parallelism but this is not demonstrated.

Another important aspect that is relegated to the appendix and not referred to in the main text is the applicability of the method to $k > 1$ which is more common in practical applications. The method seems to achieve a similar speed up also with $k = 10$ with preprocessing excluded but I would encourage the others to test with e.g. $k = 100$ as well. In addition, all experiments are done with datasets that have the same number of points and whose dimensionality is at most 200. I suspect that with at least a higher $d$ (commonly used embeddings have dimensionality in the hundreds and even thousands [2]), the speed up would be lower (due to higher preprocessing time), with a higher $N$ I'm not sure but it should be at least tested. At least in the authors' experiments, the speed up is slightly lower when the target recall is higher. Finally, the method requires tuning and additional hyperparameter $K$ which is not clear how to do with varying $N$ and $|Q|$ (the authors state that $\tau$ can be fixed as 500).

[1] Spalding-Jamieson et al. Scalable $k$-Means Clustering for Large $k$ via Seeded Approximate Nearest-Neighbor Search. arXiv:2502.06163

[2] Jääsaari et al. VIBE: Vector Index Benchmark for Embeddings. arXiv:2505.17810

**Questions:**

- How exactly does the search procedure work when $k > 1$?

- An obvious need for batch processing arises in GPU approximate nearest neighbor search. How does e.g. the method used in CAGRA [1] relate to your definition of MQO and how feasible is your approach on GPUs?

[1] Ootomo et al. CAGRA: Highly Parallel Graph Construction and Approximate Nearest Neighbor Search for GPUs. ICDE 2024.

---

### Official Review · Reviewer_bYce · 2025-11-08

**Soundness:** 3
**Presentation:** 3
**Contribution:** 2
**Rating:** 4
**Confidence:** 3

**Summary:**

### Paper Summary

This paper presents a Multiple-Query Optimization (MQO) framework for ANNS. The key idea is to accelerate batch query processing by reusing intermediate computation. Instead of using a fixed, global entry point for all queries, the framework initializes each query using the nearest neighbor result of a previously processed, spatially close query from the same batch.

To organize this process, the authors propose two algorithms: (1) MQO framework, which constructs a MST over query batch and determines the processing order accordingly; (2) Rapid query preprocessing, which partitions the query batch into a spanning forest to reduce preprocessing overhead.

The framework is evaluated on several graph-based methods and demonstrates significant speedups without requiring modifications to the underlying index structures or search algorithms.

**Strengths:**

### Paper Strengths

S1. The paper proposes a general MQO framework that can be applied on top of existing graph-based ANNS methods without modifying the underlying index.

S2. The introduction of rapid query preprocessing reduces the trade-off between optimization quality and preprocessing cost.

S3. The method is evaluated on multiple standard datasets and several state-of-the-art graph-based indexes, confirming its effectiveness and generality.

**Weaknesses:**

### Paper Weaknesses

W1. The motivation and necessity of the work are not well established. (See D1)

W2. The paper lacks discussion and comparison with existing parallel query methods and other ordered index structures. (See D2-D4)

W3. The evaluation misses experiments on smaller query batches and lacks statistics on individual query latency. (See D5, D6)

### Detailed Comments

D1. The introduction and preliminaries should better contextualize MQO within vector search. Moreover, the application example discussed in the conclusion seems inconsistent: while the proposed MQO method is designed to optimize the processing order of queries that differ greatly from each other, the cited application—RAG—produces queries that are “rewritten or augmented” from a single source and therefore tend to be close together in the embedding space. If the queries are already very similar, building a complex MST may be unnecessary; a fixed global entry point could achieve similar results. The authors should clarify why the MST is still useful in this case.

D2. The paper would be considerably strengthened by a discussion and comparison with the standard approach for batch query processing—multi-threaded parallelism [1]. Since batching is commonly used to improve throughput via parallel execution, omitting this baseline makes it difficult to assess whether the added complexity of MQO provides practical benefits.

D3. The main experiments are conducted on a single core. To better demonstrate real-world impact, the multi-threaded analysis (currently in Appendix J) should be moved to the main paper and extended with comparisons against standard parallel baselines discussed in D2.

D4. Related work on ordered index structures (e.g., KD-Tree, Z-Tree [2], CLEANN-Tree [1]) should be discussed or compared with MQO, as these methods also optimize query traversal order.

D5. The experiments mainly use very large batch sizes (100K–1M queries). Evaluating the framework on smaller, more practical batch sizes (e.g., 10, 100, or 1,000 queries) would provide a more complete understanding of its scalability and practical utility.

D6. The paper currently reports only total or average latency. In an MST-based dependency chain, queries processed later (farther from the root) may suffer from higher individual latency. The authors should include latency distribution metrics to verify that the optimization benefits all queries fairly and does not disproportionately delay some queries.


[1] ParlayANN: Scalable and Deterministic Parallel Graph-Based Approximate Nearest Neighbor Search Algorithms

[2] Parallel Nearest Neighbors in Low Dimensions with Batch Updates

**Questions:**

Please refer to weakness and detailed comments.

---

### Note · Authors · 2025-12-02

I have read and agree with the venue's withdrawal policy on behalf of myself and my co-authors.